

# Evidence-based recommendations on storing and handling specimens for analyses of insect microbiota

Tobin J. Hammer[1,2], Jacob C. Dickerson[1] and Noah Fierer[1,2]

[1] Department of Ecology and Evolutionary Biology, University of Colorado at Boulder, Boulder, CO, United States

[2] Cooperative Institute for Research in Environmental Sciences, University of Colorado at Boulder, Boulder, CO, United States

Corresponding author
Tobin J. Hammer,
tobin.hammer@colorado.edu

## ABSTRACT

Research on insect microbiota has greatly expanded over the past decade, along with a growing appreciation of the microbial contributions to insect ecology and evolution. Many of these studies use DNA sequencing to characterize the diversity and composition of insect-associated microbial communities. The choice of strategies used for specimen collection, storage, and handling could introduce biases in molecular assessments of insect microbiota, but such potential influences have not been systematically evaluated. Likewise, although it is common practice to surface sterilize insects prior to DNA extraction, it is not known if this time-consuming step has any effect on microbial community analyses. To resolve these methodological unknowns, we conducted an experiment wherein replicate individual insects of four species were stored intact for two months using five different methods—freezing, ethanol, dimethyl sulfoxide (DMSO), cetrimonium bromide (CTAB), and room-temperature storage without preservative—and then subjected to whole-specimen 16S rRNA gene sequencing to assess whether the structure of the insect-associated bacterial communities was impacted by these different storage strategies. Overall, different insect species harbored markedly distinct bacterial communities, a pattern that was highly robust to the method used to store samples. Storage method had little to no effect on assessments of microbiota composition, and the magnitude of the effect differed among the insect species examined. No single method emerged as "best," i.e., one consistently having the highest similarity in community structure to control specimens, which were not stored prior to homogenization and DNA sequencing. We also found that surface sterilization did not change bacterial community structure as compared to unsterilized insects, presumably due to the vastly greater microbial biomass inside the insect body relative to its surface. We therefore recommend that researchers can use any of the methods tested here, and base their choice according to practical considerations such as prior use, cost, and availability in the field, although we still advise that all samples within a study be handled in an identical manner when possible. We also suggest that, in large-scale molecular studies of hundreds of insect specimens, surface sterilization may not be worth the time and effort involved. This information should help researchers design sampling strategies and will facilitate cross-comparisons and meta-analyses of microbial community data obtained from insect specimens preserved in different ways.

## INTRODUCTION

Many insects are associated with microbial symbionts that play critical roles in their ecological interactions and have shaped their evolutionary history (*Dillon & Dillon, 2004*; *Engel & Moran, 2013*; *Douglas, 2015*). The structure and function of insect microbiota are increasingly studied using molecular methods such as 16S rRNA gene sequencing, but these methods are often directly adopted from environmental or human microbiota studies and have rarely been evaluated for their efficacy and accuracy when used with insect specimens. The growing field of insect-microbial symbiosis would benefit from improved validation of the sample storage and handling strategies used for studying insect microbiota.

Sample storage is one of the most important steps to consider when designing and implementing any DNA sequencing-based analysis of insect-associated microbial communities. When insect specimens are collected in the field, it is often not possible to extract DNA immediately after insect collection, and so specimens must be preserved in some way that minimizes the impact to microbial community structure. Samples are often irreplaceable, and if they are preserved improperly then they will remain useless no matter what technological advances in DNA extraction, amplification, or sequencing arise in the future (*Cary & Fierer, 2014*).

While some studies have shown that storage methodology can affect DNA recovery and amplification success from insects and/or their symbionts (*Post, Flook & Millest, 1993*; *Fukatsu, 1999*; *Mandrioli, Borsatti & Mola, 2006*; *Moreau et al., 2013*), we know of no study that has directly tested how different storage methods impact molecular assessments of insect-associated microbial community structure. Previous studies of insect microbiota have used a range of storage methods including, but not limited to, freezing (*Jones, Sanchez & Fierer, 2013*; *Hammer, McMillan & Fierer, 2014*), ethanol (*Koch et al., 2013*; *Estes et al., 2013*), RNAlater (*Campbell et al., 2004*; *Sanders et al., 2014*), and combinations thereof. We do not know how these different storage strategies may influence assessments of microbial community structure and whether some storage strategies are better suited for some insect groups than others. Furthermore, it remains poorly understood whether and to what degree any potential biases associated with sampling handling strategies may influence conclusions from meta-analyses (*Colman, Toolson & Takacs-Vesbach, 2012*) or individual studies surveying microbiota of insects stored in different ways.

Storage methods could impact insect microbiota through differential preservation of some microbial cell types (e.g., gram-positive versus gram-negative bacteria), differential penetration into insect tissues, or due to the preservative itself contaminating the specimen. Studies comparing the most commonly used storage methods on non-insect samples (e.g., soil, plant leaves, and human feces) have generally found that effects on assessments of bacterial community structure are nonexistent, or negligible in magnitude

relative to biological factors that are of interest such as interindividual or interspecific differences in microbiota (*Lauber et al., 2010*; *Rubin et al., 2013*; *U'Ren et al., 2014*; *Dominianni et al., 2014*; *Franzosa et al., 2014*).

A distinct, but related issue in processing insect specimens for molecular analysis of their microbiota is determining whether the specimens require surface sterilization prior to DNA extraction. This approach typically involves soaking specimens in dilute bleach and is often employed to remove any surface-associated microbes or contaminants from handling or storage reagents so that downstream analyses will just capture internal symbionts, such as those in the gut, bacteriomes, or reproductive structures. Despite the widespread use of surface sterilization (e.g., *Rosengaus et al., 2011*; *Hammer, McMillan & Fierer, 2014*; *Sanders et al., 2014*) and the significant amount of time it adds when extracting DNA from hundreds of specimens, its potential impact on molecular assessments of insect microbiota has not yet been evaluated. If the vast majority of microbial biomass is found inside insects, surface sterilization might have little effect on the whole-insect microbiota. Alternatively, if the procedure is too aggressive, surface sterilization could decrease the abundance of internal microbiota and introduce biases in the assessment of microbial community composition.

In this study, we asked: (i) Do storage and surface sterilization methods affect insect-associated microbial community structure, and if so, are these effects dependent on the insect species examined? and (ii) How do potential methodological biases compare in magnitude relative to the (presumably large) inter-order differences in microbiota? To address these questions, we conducted an experiment on replicate individuals from four morphologically and phylogenetically disparate insect species collected from the same locality. Specimens were stored for two months using five different treatments before DNA extraction for 16S rRNA-based community profiling. Our goal was to develop and validate a set of "best practices" for the storage and handling of samples prior to DNA-based investigations of insect microbiota.

## MATERIALS AND METHODS

During the summer of 2013, we collected adult specimens of four insect species in Boulder, Colorado, USA: the cabbage white butterfly *Pieris rapae* (Lepidoptera: Pieridae), the speckle-winged grasshopper *Arphia conspersa* (Orthoptera: Acrididae), the Mexican bean beetle *Epilachna varivestis* (Coleoptera: Coccinellidae), and the honey bee *Apis mellifera* (Hymenoptera: Apidae). We chose to collect the specimens as they would typically be collected by entomologists in order to assess the potential value of previously collected insect specimens for molecular microbial studies, and to evaluate the role of surface sterilization in removing potential contaminants introduced from human skin or collection equipment when samples are not collected or handled aseptically. For example, we used standard nets as well as manual collection (without gloves), and sacrificed insects in jars containing ethyl acetate (*Willows-Munro & Schoeman, 2014*). Insect bodies were stored intact, except butterflies from which we first removed the wings.

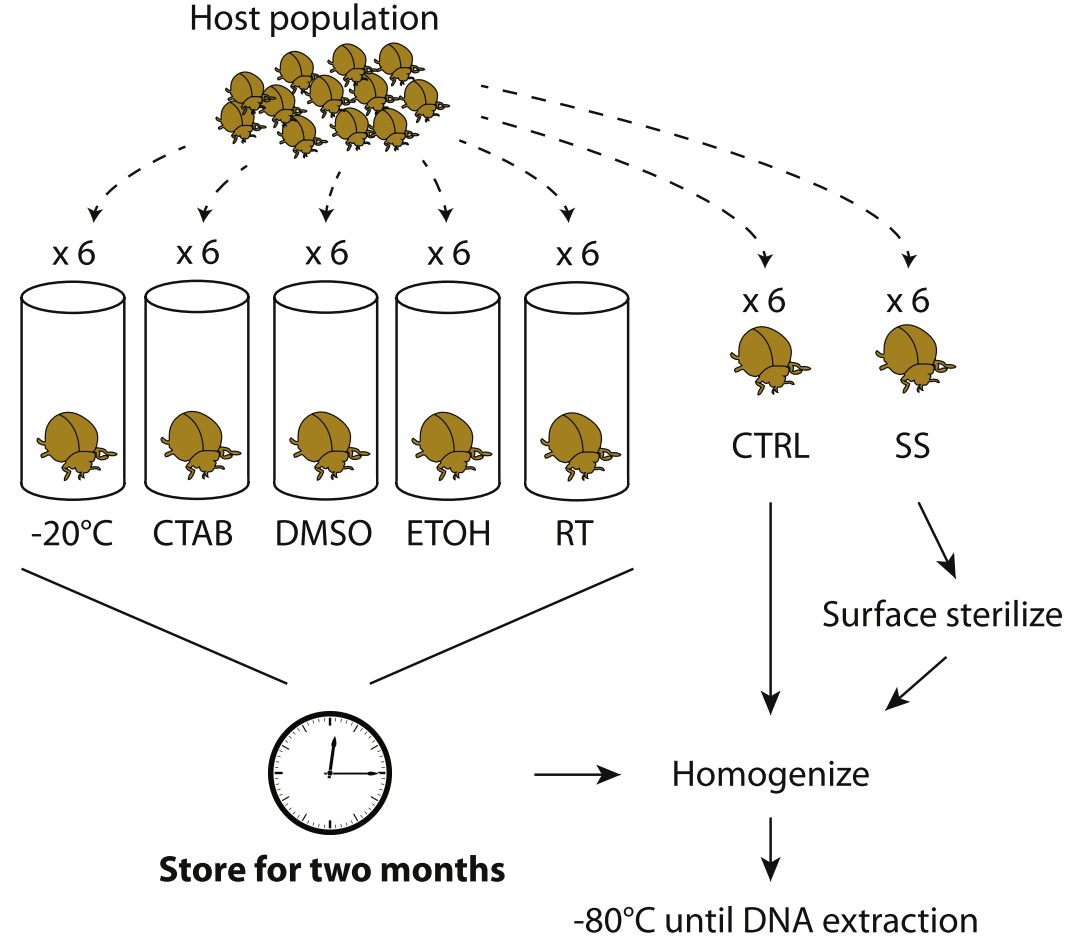

**Figure 1 Experimental design.** Overview of how insect samples were collected and processed for later whole-specimen bacterial 16S rRNA gene sequencing. Only beetles are shown, but the design was identical for the other three insect species used in the study. CTAB, cetrimonium bromide; DMSO, dimethyl sulfoxide; ETOH, 95% ethanol; RT, room temperature without preservative; CTRL, control (no storage or sterilization prior to homogenization); SS, surface sterilized (sterilization but no storage prior to homogenization).

By collecting multiple members of the same population at the same time, we aimed to minimize external sources of microbiota variability and thus maximize the potential to detect storage treatment effects. Unlike previous studies on soil or feces (e.g., *Lauber et al., 2010*; *Rubin et al., 2013*), a single insect sample cannot be equally divided into sub-samples to test multiple storage treatments; hence, our replicates are individual specimens and we expected some natural variability in microbiota among them. Six individuals were collected for each species-treatment combination, although not all specimens yielded sufficient sequence data for inclusion in downstream analyses. An overview of our experimental design is shown in Fig. 1.

To test for surface sterilization effects, one set of specimens (chosen randomly from those collected) were surface-sterilized immediately after field collection through a rinse in sterile water (Sigma-Aldrich, Seelze, Germany), a soak in 70% ethanol followed by

10% bleach for 60 s each, and a second rinse in water (similar to *Hammer, McMillan & Fierer, 2014*, though without subsequent immersion in liquid $N_2$). We then thoroughly homogenized the insects with a sterile mortar and pestle, in order to recover DNA from all potential internal symbionts (*Rubin et al., 2014*). The unsterilized control specimens were homogenized directly after field collection. All homogenized material was frozen at $-80\,°C$ until DNA extraction.

To test for effects of different storage strategies, whole insects were stored (without prior surface sterilization) over eight weeks using one of five different methods detailed below, and then homogenized and frozen at $-80\,°C$. These samples were later compared to the controls (specimens homogenized directly after field collection and frozen at $-80\,°C$ prior to DNA extraction). Although it was not possible to extract microbial DNA from freshly caught insects, we expected the controls to be the most similar to wild insect microbiota, and thus consider them to have the fewest potential biases of any of our treatments. One storage treatment simply involved maintaining dead insects in empty tubes in the laboratory, under ambient conditions (approximately 21 °C and 32% humidity). This treatment mimics how insects are often stored in museum collections, and we expected this treatment to have the strongest effect relative to the control.

The second storage method involved freezing specimens dry (i.e., no preservative) at $-20\,°C$. Freezing is commonly used and should have a lower potential for introducing contaminants than using reagents, but is often impractical in the field and makes sample transport difficult. The final three storage methods—95% ethanol, dimethyl sulfoxide (DMSO) and cetrimonium bromide (CTAB)—are all liquid preservatives that can be used without a freezer. Storage in ethanol is frequently used in insect microbiota studies, and ethanol is relatively inexpensive and easy to acquire in the field, but its flammability often makes it unsuitable for transporting specimens on planes (*Moreau et al., 2013*). DMSO and CTAB are promising nonflammable chemicals that have also been used to preserve specimens for later DNA sequencing (*Nagy, 2010*). DMSO and CTAB were salt-saturated and prepared at 20% and 2% concentrations, respectively. We note that these storage methods are sometimes used in combination—e.g., ethanol storage at $-20\,°C$—but here we evaluate them separately.

We extracted total DNA from subsamples of whole insect homogenates—which were further pulverized with an aggressive bead-beating step—using the MoBio Powersoil Kit, from which a ≈300 bp portion (V4 region) of the bacterial 16S rRNA gene was then PCR-amplified, as previously described (*Barberán et al., 2014*). The barcoded amplicons were pooled and sequenced on the Illumina MiSeq platform (*Caporaso et al., 2012*). Sequence data were quality-filtered and clustered into operational taxonomic units (OTUs) at the 97% level using the UPARSE pipeline (*Edgar, 2013*) following (*Ramirez et al., 2014*). OTU taxonomic affiliations were assigned using the RDP classifier (*Wang et al., 2007*) against the August 2013 version of the Greengenes database (*McDonald et al., 2012*), and OTUs identified as mitochondria or chloroplasts were removed. The total proportion of mitochondrial or chloroplast reads for each insect species was: bees, 0.02; grasshoppers, 0.38; beetles, 0.20; butterflies, 0.12. Finally, to standardize sequencing depth across our

dataset, we rarefied (randomly subselected) 2,000 sequence reads from each sample. Sequence data have been deposited and made publicly available on Figshare (http://dx. doi.org/10.6084/m9.figshare.1396464).

Analyses were conducted in R v. 2.13.1 (*R Core Team, 2013*), using the vegan package for multivariate analyses (*Oksanen et al., 2013*) and ggplot2 (*Wickham, 2009*) to produce plots. We quantified differences in microbial community structure between samples using the Bray-Curtis dissimilarity metric, after square-root transformation. Associations between microbiota structure and species or storage factors were tested with permutational multivariate ANOVAs (*Anderson, 2001*). The interaction between species and storage variables was included in all such tests. For multivariate analyses, all $R^2$ and $P$ values presented below were calculated using PERMANOVA, and a 0.05 significance threshold was used. Nonparametric Mann–Whitney tests were used to assess differences in the relative abundances of particular taxa between sample treatments and controls with analyses focused on those genera that had median relative abundances $\geq 1\%$ in at least one sample type, applying a false discovery rate correction to account for multiple comparisons.

## RESULTS AND DISCUSSION

### Storage effects

The four insect species we examined harbored distinct bacterial communities (Table S1), and the differences in community structure between insect species were far larger than any differences between storage conditions within individual insect species. In a model including insect species, storage treatment, and the interaction between the two, the effect of species was very strong ($R^2 = 0.866$, $P = 0.001$), as is evident from an unconstrained ordination of all the data where the samples cluster according to insect species they belonged to, and not to the method used to store them (Fig. 2). Overall, we could not reject the null hypothesis of no difference in bacterial community structure among the storage treatments ($R^2 = 0.009$, $P = 0.058$). However, there was a significant interaction, indicating that the strength of a storage effect depended upon the insect species in question ($R^2 = 0.035$, $P = 0.001$).

When each insect species was analyzed separately, each of the four insect species exhibited statistically significant effects of storage method on community structure ($R^2 = 0.286$–$0.357$, $P = 0.001$–$0.019$), although the magnitude and direction of the effect varied between them (Fig. S1). Thus, storage methods can have effects on insect-associated microbial community structure, but these effects vary by the insect, and are much smaller than species differences. In other words, storing the insect samples in six different ways, including leaving some in empty vials at room temperature for two months, did not hinder our ability to detect species effects, which are often of biological interest (e.g., *Colman, Toolson & Takacs-Vesbach, 2012*; *Jones, Sanchez & Fierer, 2013*; *Yun et al., 2014*). Our results parallel previous studies on plant, soil, and human fecal microbiota, which did not find strong effects of commonly used storage methods (*Lauber et al., 2010*; *U'Ren et al., 2014*; *Dominianni et al., 2014*; *Franzosa et al., 2014*). At least for the storage methods tested here, researchers that want to compare bacterial communities across distinct insect taxa using

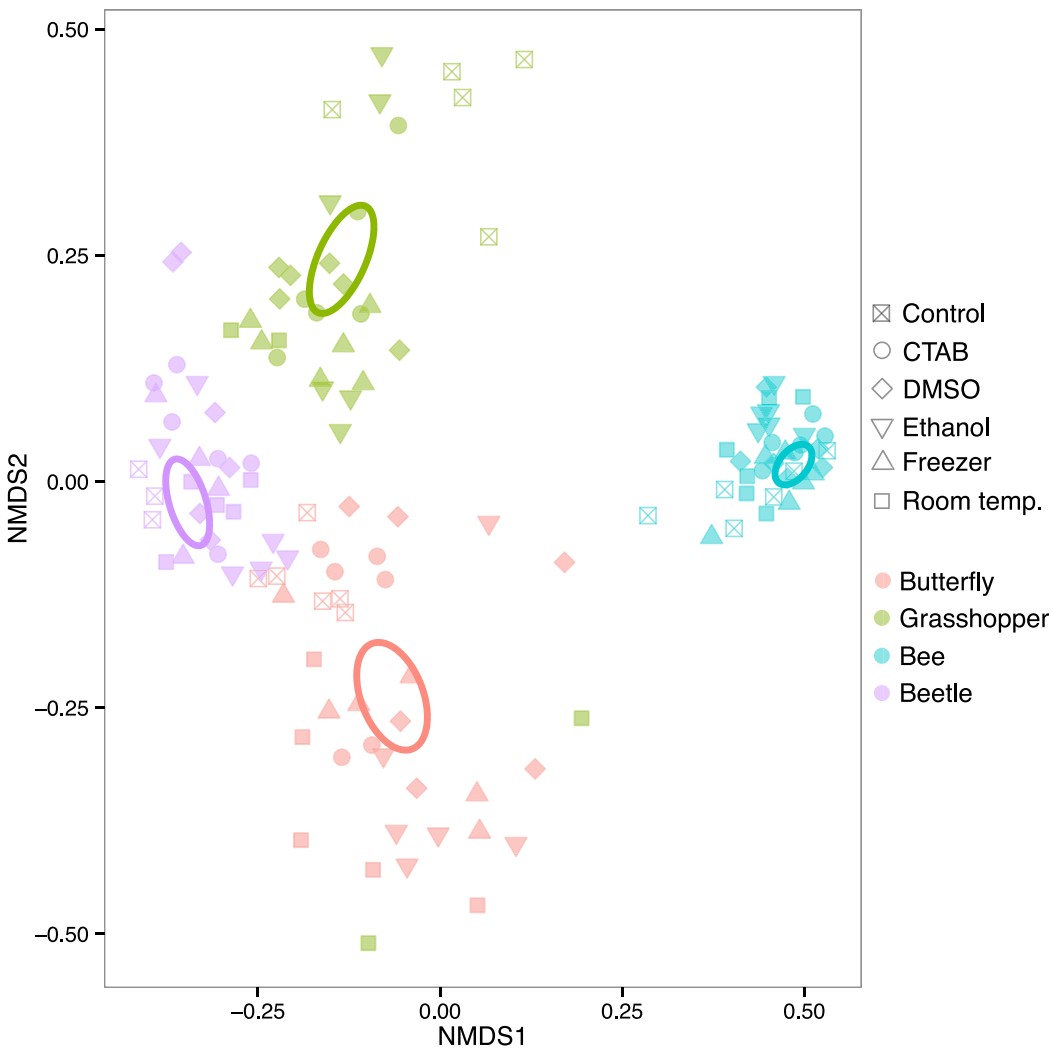

**Figure 2 Ordination of insect microbiota.** Nonmetric multidimensional scaling ordination showing how insect-associated bacterial communities cluster by host insect species (colors) versus specimen storage method (symbols). Ellipses represent 95% confidence intervals around species group centroids.

samples stored under different conditions—e.g., with sample sets compiled from multiple collectors, or in meta-analyses—may not need to worry, at least when the insect species harbor distinct bacterial communities. However, as our focal insects belong to different taxonomic orders with distinct ecological and physiological attributes, specimen storage may be of more concern when investigators are studying more closely related species. Future research into potential storage biases on microbiota of congeneric insects, for example, would be useful in guiding research at narrower taxonomic scales.

Given that we did find a significant effect of storage conditions on our assessments of bacterial community composition within each of the four insect species examined, we recommend that, when possible, the same method be used to store all specimens within a study. In order to help guide researchers in their choice of storage method, we examined whether some storage methods were consistently better than others, i.e., whether some

**Table 1 Comparison of storage methods.** Results from pairwise PERMANOVA tests conducted for samples from each storage method with the controls. Methods are ranked by increasing percentage of variance explained by the storage term. Ordination plots that correspond to these tests are shown in Fig. S2.

| Storage method (versus control) | Model term | $R^2$ | *P* values |
|---|---|---|---|
| Freezer | Species | 0.849 | 0.001* |
| | Storage | 0.003 | 0.285 |
| | Species × Storage | 0.047 | 0.002* |
| CTAB | Species | 0.862 | 0.001* |
| | Storage | 0.005 | 0.149 |
| | Species × Storage | 0.030 | 0.010* |
| Ethanol | Species | 0.820 | 0.001* |
| | Storage | 0.007 | 0.168 |
| | Species × Storage | 0.035 | 0.016* |
| DMSO | Species | 0.839 | 0.001* |
| | Storage | 0.011 | 0.040* |
| | Species × Storage | 0.046 | 0.001* |
| Room temperature | Species | 0.809 | 0.001* |
| | Storage | 0.017 | 0.041* |
| | Species × storage | 0.049 | 0.004* |

**Notes.**

Significant effects ($P < 0.05$) are indicated by asterisks.

introduced fewer biases in bacterial community analyses compared to the control. To do so, we tested for storage and species effects on bacterial community structure for each method versus the control. Only the room-temperature unpreserved treatment and DMSO had a (marginally) significant main storage effect, although there was an interaction between species and storage effects in each case (Table 1 and Fig. S2). Further, the storage effect consistently explained only a small proportion of variation in community structure, especially relative to the species effect (Table 1), which was large and easily visible in ordinations regardless of the storage method used (as mentioned above) (Fig. S2). Ranked by mean Bray-Curtis dissimilarity between each storage method and the control, there was no single 'best' method across all four insect species, and often multiple methods were not substantially distinct from one another in the extent to which the bacterial communities differed from those found in control specimens (Fig. 3). While dependent on the question of interest, we can broadly recommend any of the storage methods tested here, although room-temperature storage without preservative should be avoided when possible.

As these analyses were conducted on the structure of the entire bacterial community in each specimen, we also wanted to determine whether the different storage treatments altered the relative abundances of individual, dominant bacterial genera, and whether these effects were consistent across insect species. We found that genus-level relative abundances were generally not more similar within- than between-treatments, nor markedly different between storage treatments and the control besides natural interindividual variation (Fig. 4). Furthermore, specific genera that differed significantly between a storage method and the control were mostly concentrated in the room temperature treatment, and

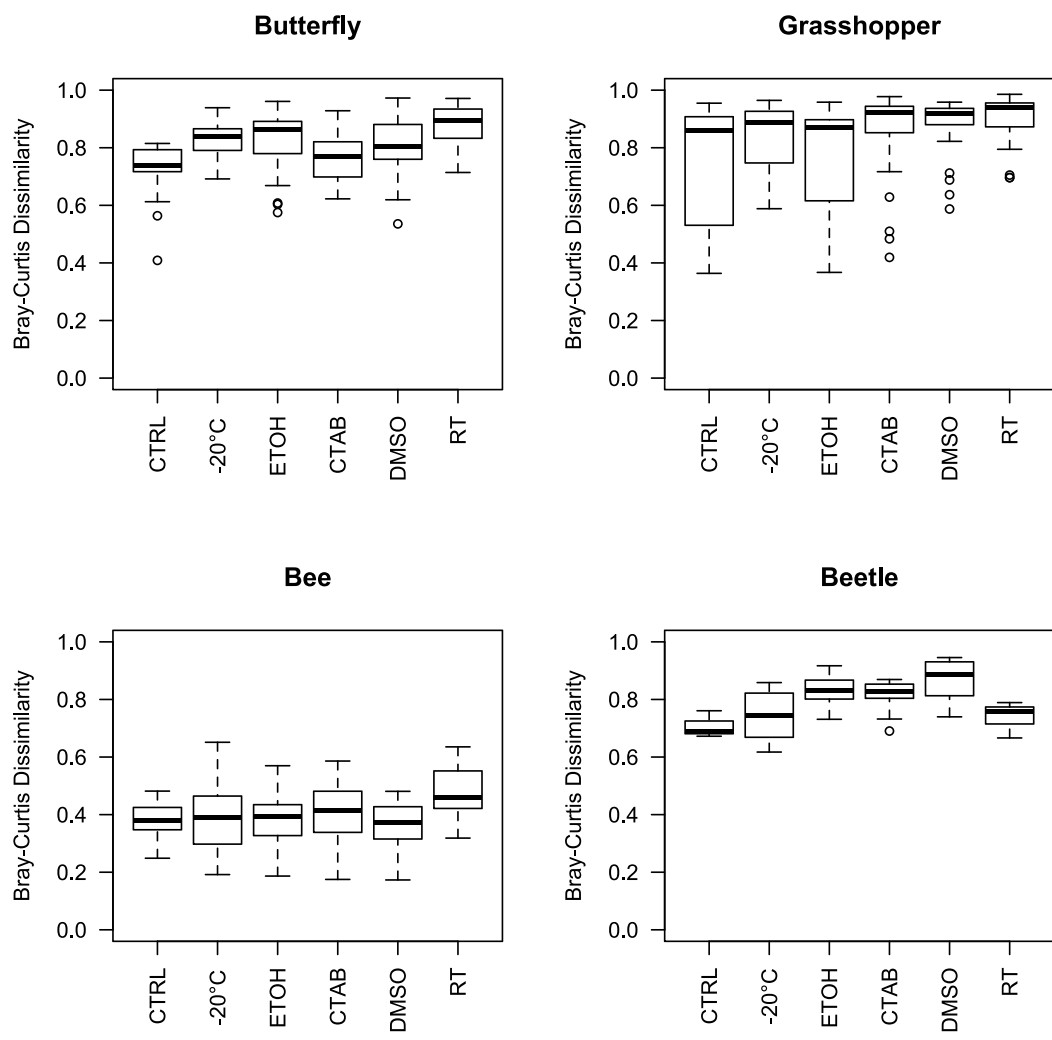

**Figure 3 Beta diversity among insect species and storage methods.** Boxplots of Bray-Curtis dissimilarities between samples from each storage method and the controls, separated by the four insect species tested. Higher values indicate that the storage method had larger effects on bacterial community structure relative to the control. Note that, due to interindividual variability in the microbiota, there was variation even among the control specimens. CTAB, cetrimonium bromide; DMSO, dimethyl sulfoxide; ETOH, 95% ethanol; RT, room temperature without preservative; CTRL, control.

none differed consistently across insect species (Table S2). This finding suggests that storage conditions are likely to exert their effects by changing the abundance of resident community members, and not by introducing new 'contaminant' taxa.

Interestingly, although honey bees had the lowest interindividual heterogeneity of the four insects tested (indicated by Figs. 3 and 4), which should have increased our ability to detect storage-induced effects, they were overall less affected by storage (excepting the room temperature treatment, where *Lactobacillus* was nearly absent (Fig. 4, Table S2)) than the other insects (Fig. 3). This finding may result from the extremely dense communities inhabiting honey bee guts (possibly reaching ca. $10^7$ bacterial colony forming units (CFUs) per bee, (*Kwong et al., 2014*)), which may buffer them from relatively

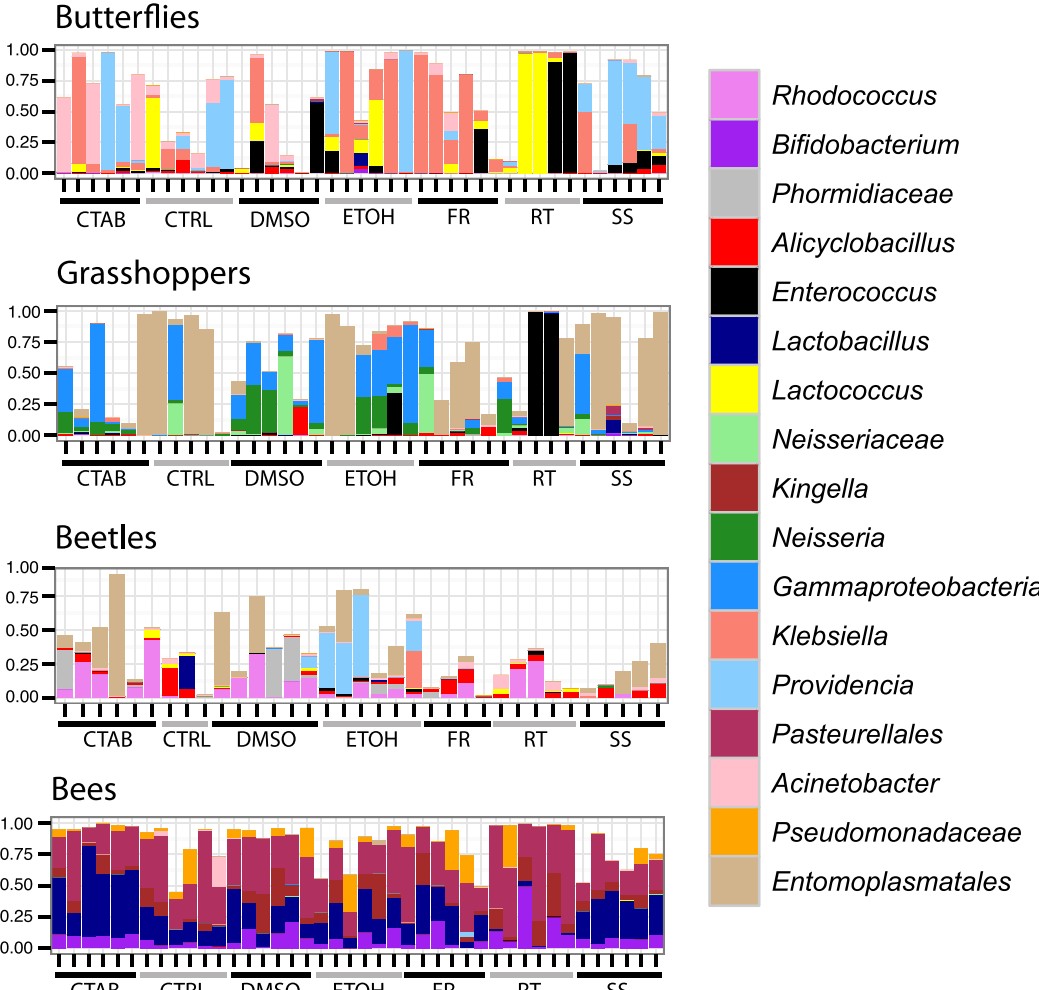

**Figure 4 Composition of bacterial communities in study insects.** Relative abundances of dominant genera for each individual specimen analyzed in the study. Seventeen genera are displayed, representing the five most abundant from each insect species (three genera were in the top five of more than one insect species). In cases where there was no genus-level identification based on the Greengenes taxonomy, the lowest-level classification is given. Blank areas show the proportion of the community for each sample that does not belong to these dominant genera. CTAB, cetrimonium bromide; DMSO, dimethyl sulfoxide; ETOH, 95% ethanol; RT, room temperature without preservative; FR, freezer at −20 °C; CTRL, control; SS, surface-sterilized.

low-abundance contaminants or minor alterations to resident populations. In contrast, bean beetles, where the apparent storage effects were stronger (Fig. 3), have been reported to contain only 600–1,750 bacterial CFUs per beetle (*Taylor, 1985*). Bacterial cell counts have not been conducted in *A. conspersa* and *P. rapae* adults, but we found substantial interindividual variation in the composition of dominant taxa for all three non-bee species, even within control and treatment groups (Fig. 4). In general, variation in both microbial cell numbers and intra-specific variability in bacterial community composition may explain the observed interactions between storage method and insect species. When feasible, quantifying microbial abundances would be a valuable complement, as storage

biases might be less of a concern for molecular investigations of insect taxa harboring particularly large numbers of microbes.

## Surface sterilization effects

A second aim of our study was to determine whether surface sterilization affected insect-associated microbiota. As compared with control specimens (which were not surface sterilized), there was no effect of surface sterilization on insect-associated microbial community structure ($R^2 = 0.004$, $P = 0.344$), and no significant interaction between species and sterilization ($R^2 = 0.022$, $P = 0.072$). Again, samples clustered strongly by species ($R^2 = 0.846$, $P = 0.001$), regardless of whether or not they were surface sterilized (Fig. S3), although there was a slight differentiation between treatments within individual species (excepting honey bees). Likewise, the relative abundances of individual genera were generally similar between sterilized and control specimens (Fig. 4).

While sterilization is usually intended to remove surface contaminants (including those derived from handling the specimen) or microbes typically present on the cuticle, we do not know if these surface-associated microbes are normally abundant enough to be detected in molecular surveys of whole-insect microbiota. In our study, where all insects were collected without gloves, we found that bacteria known to be common on human hands (including *Staphylococcus*, *Corynebacterium*, and *Propionibacterium*; *Fierer et al., 2008*) were extremely rare or absent from nearly all of our samples. Only *Staphylococcus* and *Corynebacterium* were detected and only in control *P. rapae* insects (at median proportions of 0.0005 and 0.001, respectively). In light of the general lack of a strong surface sterilization effect at the community level, in tandem with the lack of abundant human-derived contaminants even in control specimens, we argue that surface sterilization may not be worth the time and effort required. Presumably, the high microbial biomass inside many insects (such as in the gut or bacteriocytes)—relative to microbial colonizers or contaminants on the insect cuticle—overwhelms any potential effect of sterilizing the insect surface prior to microbiota analyses. Although our sterilization protocol was similar to or more aggressive than protocols used in previous studies (e.g., *Jones, Sanchez & Fierer, 2013*; *Hammer, McMillan & Fierer, 2014*; *Sanders et al., 2014*), it remains possible that modifications such as a longer soak duration could lead to detectable effects on overall microbiota. However, as we noted earlier, this may have the unintended effect of sterilizing the internal communities that are typically of interest.

## Conclusions

We recommend omitting surface sterilization from insect microbiota studies, and suggest that any of the storage methods tested here—with the possible exception of room temperature storage without preservative—can be safely used for up to two months if the researcher is not seeking subtle biological patterns that could be obscured by minor storage effects. Each method resolved species differences, and produced reasonably consistent estimates of community structure. Methods may thus be chosen based on practical considerations, such as price, availability, ease of preparation, and travel logistics. For example, CTAB may be a good choice for overseas fieldwork when electricity or dry ice is

not available, as it can be maintained at room temperature and is nonflammable. We note, however, that when planning insect collections for microbiota analysis, it is still advisable to standardize methods across all of the specimens.

In some cases, researchers are unable to choose how insects are stored in advance, but this should not necessarily limit the utility of those samples for microbial analyses. For example, samples may have already been collected and stored by entomologists for non-microbial purposes (such as morphological analyses). Our data indicate that standard entomological collection methods, where specimens are not handled aseptically, are sufficient for capturing *in situ* community structure and biological patterns, and that surface sterilization does not appear to be necessary for microbial studies. Although we do not know how different storage approaches may influence microbial analyses beyond the two month window examined here, we suggest that if storage effects are weak or nonexistent after two months, it is unlikely that they will pose a problem after longer periods of time. However, until longer-term storage effects are explicitly tested, we recommend extracting DNA from specimens within two months of storage when possible.

While further investigations into storage effects on insect microbiota would be useful—particularly tests that include combinations of various methods—we anticipate that these findings will allow existing insect samples to be used for microbial DNA sequencing, enable comparative studies that include specimens collected in different ways, and help guide the design and standardization of methods in the rapidly growing field of insect-microbe symbiosis.

## ACKNOWLEDGEMENTS

We thank Cesar Nufio, Chelsea Cook, and Megan Blanchard for their assistance collecting insects, and Jessica Henley for her help with DNA sequencing.

### Funding

T.J.H. is supported by a Graduate Research Fellowship from the National Science Foundation. The funders had no role in study design, data collection and analysis, decision to publish, or preparation of the manuscript.

### Grant Disclosures

The following grant information was disclosed by the authors:
National Science Foundation.

### Competing Interests

The authors declare there are no competing interests.

### Author Contributions

- Tobin J. Hammer conceived and designed the experiments, performed the experiments, analyzed the data, wrote the paper, prepared figures and/or tables, reviewed drafts of the paper.

- Jacob C. Dickerson performed the experiments, reviewed drafts of the paper.
- Noah Fierer conceived and designed the experiments, analyzed the data, wrote the paper, reviewed drafts of the paper.

## DNA Deposition

The following information was supplied regarding the deposition of DNA sequences:
Figshare (DOI 10.6084/m9.figshare.1396464).

## Supplemental Information

Supplemental information for this article can be found online at http://dx.doi.org/10.7717/peerj.1190#supplemental-information.

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
