# Peer review of "Evidence-based recommendations on storing and handling specimens for analyses of insect microbiota"

_PeerJ, doi:10.7717/peerj.1190_

## Round 0.1 · original submission · Minor Revisions

As you can see, all three reviewers have made very positive comments on your paper, and have suggested only relatively minor revisions.

However, two of the reviewers comments raise some important points that will need to be addressed before the paper can be accepted for publication. Therefore, if you agree to revise the paper, I would ask you to respond to each of the reviewers' comments, and explain clearly, in the usual way, how the paper has been revised to address in particular the issues raised by Reviewers 1 and 3.

Reviewer 1 ·

Basic reporting

No Comments

Experimental design

No Comments

Validity of the findings

No Comments

Additional comments

- The largest contribution of this study is that excluding surface sterilization does not seem to have any effect in biasing insect associated bacterial community recovery.
- Although I support the common use of term “microbiome” to include the identification host-associated microbes, some in the scientific community have insisted that this term by definition should be reserved for studies that show gene content and function of resident core microbes. As you have used this term in the first sense, you may want to define your use in this manuscript as only for taxonomic identification.
- It is not clear from the methods section whether the DNA extraction protocol of Rubin et al. (MicrobiologyOpen 2014; 3(6): 910–921) was followed as this study demonstrated that without a homogenization step you may miss much of the insect-associated bacterial community composition.
- I would argue that another reason people have been surface sterilizing specimens (lines 118-120) is to remove external bacteria from the host’s environment when studies have been interested in endosymbionts.
- The Methods need additional information regarding the specific analyses that were implemented. For example in the first paragraph of the Results and Discussion the authors refer to “a model” investigating the interaction among variables, but this was not presented in the Methods. Also, how was the P-value calculated (I am assuming PERMANOVA since this appears in the figure legend of Table 1 and some of the Supplementary figures, but this should be included in the methods)? Also, why where OTUs represented by single sequences not excluded (Table S1)?
- Figure S3 should be included in the main text as the identity of the bacterial community members is of general interest.
- Please include information about which categories in Figure 3 are significantly different for each insect and include information in the methods regarding how these tests were implemented.

Minor Revisions:
- Figure 1: Add “bacterial” to 16S rRNA to be clear you are targeting this gene in the bacteria and not the hosts as this gene is sometimes also sequenced in insects.
- Line 161: Replace “microbial” with “bacterial” as the 16S rRNA primers only target bacteria and not other microbes.
- What percentage of the reads were from mitochondria and chloroplasts? Please add this information to your Results for each taxonomic group. Are there any predictable patterns? Are insects that feed directly on plants in the adult life stage more likely to have higher contamination with chloroplast DNA?
- Table 1: It would be helpful to include asterisks next to the P-values (I am assuming this is what they are, although this column is only labeled “P” and there is no information in the methods about how significance was determined).
- Figure S3: Seventeen should be spelled out since it is the first word of the sentence. Also, why is “top five” in quotes? Why not just explain what this means?

·

Basic reporting

No comments

Experimental design

No comments

Validity of the findings

No comments

Additional comments

This article is clear, concise, and basically ready for publication. I only have two thoughts on how to broaden its scope.

First, the paper's findings will be a welcome resource for the insect microbiota community. The work has carefully taken into account effects of species, extraction method, etc on microbial community assembly. I wondered how sensitive the overall results are to OTU cutoff level. While not necessary for publication, could the authors consider some basic analyses at alternative OTU cutoff levels (i.e., 99%) in order to broaden the scope and utility of the paper for the community.

Second, the few species analyzed are fairly divergent. It seems probable that the sensitivity of extraction method may be more important when investigators are studying closely-related insect species in the same genera. In this regard, a note of caution (perhaps I missed it) is welcome for the community at large.

Reviewer 3 ·

Basic reporting

All seems fine.

Experimental design

All seem up to standards.

Validity of the findings

Analyses seem sound, and data are available on Figshare.

See general comments for discussion about interpretation and recommendations.

Additional comments

General comments:

This manuscript is a really nice contribution to a field that is sorely in need of more rigorous methodological inquiry. The nicely balanced design explores the effects of just about every frequently-used sample storage methodology on bacterial 16S amplicon sequence diversity, as well as the frequently-used (but dubiously justified) practice of bleach-based ‘surface sterilization.’ The data presented will be extremely useful for calibrating researchers’ intuitions. The manuscript is clear and well-written, and the data are presented in a straightforward and useful manner.

I do have some qualms about the interpretation and overall message of the manuscript, which is that storage methodology and surface sterilization are relatively inconsequential for the description of insect microbiota from 16S data (‘Storage method had little to no effect on assessments of microbiome composition’). The authors make this argument from the relatively small effect sizes of these factors relative to between-species differences in their multivariate model (Table 1). While strictly accurate, I find this conclusion to be misleading: different storage methods actually appear to result in quite substantial differences in the described bacterial community; it’s just that these differences are small relative to the enormous differences between species. In this case, the ‘species’ comparison is actually between insects of four different orders with tremendously different ecologies and physiologies. Is this really the relevant comparison for most readers?

Based on the data presented, I find it plausible that the storage effects observed here might actually be large relative to differences among species from the same genus or family for some insect lineages, which may actually be a more relevant comparison for most researchers. The authors indeed readily acknowledge this (‘Given that we did find a significant effect of storage conditions on our assessments of bacterial community composition within each of the four insect species examined, we recommend that, when possible, the same method be used to store all specimens within a study.’), so I don’t think any radical changes are needed – just a little bit more exploration of the issue as it is likely to be relevant to other questions.

Overall, though, I find this to be an excellent and timely (really, overdue) contribution.


Specific comments:

35: It might be useful to clarify what ‘control’ was in the abstract.

103: ‘Interspecific’ doesn’t sound quite right to me, given that the differences being measured here are really between orders. In many cases, the apparent differences in microbiota between species are quite subtle.

135: If homogenization was performed on LN2 (as in the referenced protocol), it should also be noted here.

141: Is the only difference between ‘controls’ and the -20° treatment those 60° C? Or that the controls were collected, flash frozen & homogenized in LN2, and the homogenate stored at -80° vs the whole organism being stored at -20° first? It would be nice to a have a more easily-parsed explanation of what ‘control’ really means here.

146: It would be useful to report at least a sense of what those ambient conditions were -- I suspect that if I left insects in vials for two months in more humid conditions the change would be more substantial.

175: It would be very useful to also be able to see data for another measure of beta diversity. For instance, I would expect the change in the room-temperature honeybee samples to be much more exaggerated using a weighted phylogenetic measure.

192: I don’t think this sentence is an accurate interpretation of the statistical result. My reading is that there was only a 5.8% chance that the differences in centroid observed between treatments could have been observed if treatment had no net effect. You can’t reject the null, but you do come pretty damn close.

225: Again, this isn’t the recommendation that I feel like I’d make to someone based on these results -- it seems like it’s strongly dependent on the question.

247: Are CFUs the most appropriate measure of abundance here? Even if not, I think invoking absolute abundance is really important, and could bear some additional attention in the manuscript as well. Wouldn’t a clear recommendation be to also look into measurement of absolute abundance in your samples, as well?

260: Can this be right? Figure S4 shows a pretty consistent trend for non-bee insects along NMDS axis 2. Is that really not showing up as significant in your PERMANOVA? The results (along with what little can be gleaned from Figure S3) seem consistent with the idea that there are some bugs that you get ride of by surface sterilizing.

276: Is this really a safe presumption?

Figure 3: I don’t think SEM is an appropriate measure of spread here, given that these are non-independent pairwise comparisons (SEM underrepresents the spread). Boxplots would give a much better intuition of the distribution. It would also be very helpful to have within-group distributions for each of preservation methods.

Figure S1: Possible to ditch some of the wasted space on these plots? This is actually an important figure (as you’re not also compressing treatment variation to see between-order variation) and it would be helpful to see it less cramped.

Figure S2: Similarly, for charts with two subcategories, the default shapes are hard to distinguish. It would be helpful to specify open shapes for one category and filled for the other.

---

## Round 0.2 · accepted · Accept

It is clear to me that you have revised the paper substantially according to the reviewers' comments, which, I agree, has improved it. Where you have argued against specific (minor) changes, I find your rebuttals justified, and am happy to accept the revised paper.